# Passive electroculture using copper rods does not improve yield in home container vegetable gardening

Mya Chier[1], Aidan Oakey[1], Michelle L. Budny[1], Nathan P. Lemoine [1,2*]

1 Department of Biological Sciences, Marquette University, Milwaukee, Wisconsin United States of America, 2 Department of Zoology, Milwaukee Public Museum, Milwaukee, Wisconsin United States of America

* nathan.lemoine@marquette.edu

⊙ OPEN ACCESS

## Abstract

Electroculture, or the practice of applying electric fields or current to plants, has been explored for nearly three hundred years. Recently, home gardeners on social media have adopted the term electroculture to describe inserting copper-wrapped dowels into root soil as a cost-effective means of improving crop yield in small garden spaces. Given the renewed interest, big box stores have begun stocking copper-wrapped dowels as a means of improving plant growth in urban container gardens, yet whether such passive electroculture is sufficiently beneficial to plant growth to justify the cost and materials, particularly of copper, remains debated. It is likely that copper rods produce too little voltage to affect plant physiology. In this study, we tested the hypotheses that 1) inserting copper-wrapped dowel rods into the soil will not improve plant growth, photosynthesis, or yield, and 2) if copper-wrapped dowel rods improve plant growth, it is due to copper fertilization rather than electrical conductance. We tested these hypotheses on two leafy green vegetables, mustard greens and kale, and two root vegetables, beets and turnips, to determine if plant life history was an important factor in the efficacy of passive electroculture. We found no consistent evidence that passive electroculture is beneficial to crop growth or yield in container gardens. Although we documented statistically significant effects of buried copper on the above- and belowground biomass of turnips, it is unlikely that improved turnip yield was due to copper fertilization because plants grown with exposed copper rods did not show the same effect. While crop production could potentially be enhanced by the application of active electrical fields, the voltages required exceed what is produced by copper-wrapped wooden dowels. We therefore suggest that both the production and purchase of such products would waste both financial and natural resources.

**Data availability statement:** Data are available within the Figshare database: 10.6084/m9.figshare.28752290.

**Funding:** US National Science Foundation DEB 1941390. The funders took no part in this study.

**Competing interests:** The authors have declared that no competing interests exist.

## Introduction

Electroculture, or the practice of applying electric fields or current to plants [1], has been explored for nearly three hundred years. In the 18th century, Jean-Antoine Nollet reported that applying electrical current improved plant growth [2]. In the late 19th century, Selim Lemström reported that electrical stimulus improved the growth of potatoes, celery, and carrots by 40–70% over a two month period [3]. Throughout the early- and mid-20th century, electroculture received sporadic attention as a potential means of improving agricultural yield [4–7], although the practice has never been widely adopted in commercial settings. Recently, home gardeners on social media have adopted the term electroculture to describe inserting copper-wrapped dowels into root soil as a cost-effective, passive means of improving crop yield in small garden spaces [8]. Given the renewed interest, big box stores have begun stocking copper-wrapped dowels as a means of improving plant growth in urban container gardens, yet whether such passive electroculture is sufficiently beneficial to plant growth to justify the cost and materials, particularly of copper, remains debated.

Electrical stimulation could theoretically improve photosynthesis and thereby enhance crop production by mimicking electrical signaling pathways. For example, electrical signaling appears to be among the first rapid signals to stress, which ultimately helps maintain net $CO_2$ uptake [9]. During temperature stress, electrical signals help maintain stomatal $CO_2$ uptake, potentially preventing stomatal closure caused by dropping foliar water potentials [10]. Electrical signals also propagate from the roots up to the leaves, serving as a potential signaling mechanism to increase photosynthetic activity in the presence of increased nutrient supply [11]. These lines of evidence suggest that an applied electrical current might potentially aid crop production by maintaining $CO_2$ uptake, particularly during stressful events. However, electrical stimulus can also serve to depress photosynthesis as a response to foliar damage, similar to the effect of mechanical wounding on leaves [12,13]. In such negative feedbacks, electrical stimuli might serve to repress photosynthesis and crop yield. Indeed, strong electrical fields and currents have a multitude of effects, such as increased yield and germination rate, decreased yield, increased chemical defense production (similar to defense induction by wounding), or decreased or increased germination times [14]. The effect of electrical stimulus appears to depend on both the plant species and the type and magnitude of electrical current applied [14], and whether copper dowels passively provide enough electricity to promote crop growth remains untested.

It is also possible that plant physiology and performance can be altered by directly by the copper coil inserted into the soil. At high concentrations, copper becomes photo-inhibitory and forms the base metal of many herbicides. Experiments have repeatedly demonstrated that excess copper can inhibit photosynthesis [15,16]. Interestingly, photosynthesis appears to be inhibited during carbon fixation, rather than via stomatal limitation [16]. At low concentrations, however, copper amendments can improve yield. Copper fertilization increased wheat photosynthesis and yield by over 100% in copper-limited soils of Canada and India and also stimulated photosynthesis of hydroponically grown rice [17–19]. Small amounts of copper contained in

fungicide applications also resulted in a temporary increase in $CO_2$ assimilation of hops [20]. Yet no controlled study exists to determine whether the benefits of passive electroculture are due to copper fertilization rather than electrical stimulation of plants.

In this study, we tested two hypotheses:

**H1:** Inserting copper-wrapped dowel rods into the soil will not improve plant growth, photosynthesis, or yield. Although electrical stimulation can improve plant performance, it is unlikely that a copper dowel passively transmits enough electricity into the soil aid plant health. We therefore predicted that $CO_2$ assimilation, growth rate, and biomass should be similar between plants grown with and without copper-wrapped dowels.

**H2:** If copper-wrapped dowel rods improve plant growth, it is due to copper fertilization rather than electrical conductance. If true, we predicted that burying copper-wrapped dowel rods severed at the soil surface would enhance plant performance similarly to copper-wrapped dowels that extend above the soil surface.

We tested these hypotheses on two leafy green vegetables and two root vegetables to determine if plant life history was an important factor in the efficacy of passive electroculture.

## Methods

### Plant germination

We selected four common garden species for our study: two leafy green varieties, mustard greens (*Brassica* spp.) and white Russian kale (*Brassica* spp.), and two root vegetables, turnips (*Brassica* spp.) and beets (*Beta* spp.). Seeds germinated in an environmental growth chamber (Conviron GEN1000; Winnipeg, Canada) set to 24°C, 85% relative humidity, and 15:9 hour photoperiod. After four weeks we transplanted seedlings into individual one-gallon pots with garden potting soil and moved the pots to a lab bench under LED grow lights. Ambient room temperature stayed near 25°C. We measured soil moisture weekly with a TDS Field Scout soil moisture meter and watered plants to maintain 20–30% soil moisture.

### Copper stimulation of plant growth

We tested the effects of passive copper electroculture by randomly assigning individual plants to one of three treatments: a control group with no added copper, exposed copper wire, and buried copper wire ($n = 10$ plants per treatment, per species). The exposed copper wire treatment consisted of a 40 cm dowel rod coiled with copper wire and inserted 10 cm into the soil. To control for potential copper fertilization effects, we buried 10 cm dowel rods coiled with copper wire into the pots completely covered by the soil. Each week we measured plant height from the soil surface to the highest point on the stem. After eight weeks, all aboveground biomass was clipped and dried to a constant weight, and belowground biomass of beets and turnips was harvested, dried and weighed. We did not measure belowground biomass for mustard greens and kale since their roots were too fine to separate from the soil without substantial root damage or loss.

### Plant physiology measurements

We measured leaf greenness weekly using a non-destructive SPAD meter (Konica-Minolta) as a proxy for chlorophyll content, which requires destructive sampling. Net $CO_2$ assimilation ($A_{net}$) and stomatal conductance to water vapor ($g_{sw}$) were measured using an LI-6800 portable photosynthesis analyzer (Li-COR, Lincoln, Nebraska, USA). We measured $A_{net}$ and $g_{sw}$ on young fully expanded leaves with no signs of damage under the following conditions: flow rate = 400 µmol s$^{-1}$, $CO_2$ reference = 420 µmol mol$^{-1}$, relative humidity = 50%, light = 1500 µmol m$^{-2}$ s$^{-1}$, fan speed = 10,000 rpm, leaf temperature = ambient conditions. We enabled dynamic equations on the LI-6800 to calculate $A_{net}$ [21]. Leaves were allowed to acclimate until $A_{net}$ was stable (maximum of 2 minutes) before logging data.

## Statistical analyses

We tested treatment effects of copper wire on $A_{net}$, $g_{sw}$, height, and SPAD using a repeated measures ANOVA. We compared the likelihood ratio of the basic model (week) to the treatment model (week + treatment), and the treatment model to the interaction model (week + treatment + week:treatment). If a likelihood ratio was significant ($p < 0.05$), we conducted a Tukey pairwise *post-hoc* test to determine which groups differed. To reduce the number of post-hoc tests and therefore the severity of the *p*-value corrections, we only compared treatments within a given week and not across weeks. We assessed water use efficiency (WUE) by log-transforming $A_{net}$ and $g_{sw}$ and performing an ANCOVA to test the slopes (i.e., $CO_2$ uptake/ $H_2O$ loss) by treatment.

To find the treatment effect of copper on aboveground biomass of each species, and belowground biomass of turnips and beets, we performed ANOVAs of each treatment. If the ANOVA was significant, we used a Tukey pairwise test to find which treatments had a significant interaction.

## Results

### Physiology

Neither exposed nor buried copper affected $A_{net}$ of kale, mustard greens, or turnips, while beets had a significant an interaction between copper treatment and week (Fig 1, Table 1). In week 6, beet plants with exposed copper had lower $A_{net}$ than both control plants ($p < 0.001$) and plants with buried copper ($p < 0.001$), and control plants did not differ from plants with buried copper ($p = 0.63$, Fig 1C). However in week 8, plants with buried copper had lower $A_{net}$ than the control plants ($p = 0.02$) and plants with exposed copper ($p = 0.04$). We found no effect of copper treatment on $g_{sw}$ for mustard greens, beets, or turnips. Kale, on the other hand, did show a treatment effect but not for the interaction with week (Table 1). Kale plants with buried copper had 13% higher $g_{sw}$ than the control plants ($p = 0.02$) and 18% higher $g_{sw}$ than plants with exposed copper ($p < 0.01$) over the course of the experiment (Table 1, Fig 2).

We found no interaction of copper treatment and WUE for kale, mustard greens, or beets (Fig 3, Table 2). Initially we did find an interaction of WUE in turnips. The slope of the control plants was shallower than the slope for exposed and buried, suggesting that control plants had lower WUE than either copper treatment. However, the shallow slope was driven by the inclusion of one outlier that, when excluded, removed the effect of copper treatment on WUE for control plants (Fig 3, Table 2).

### Height and SPAD

Neither treatment nor the interaction of treatment and week was significant for height or SPAD for any species (Table 1).

### Biomass

Copper treatments affected aboveground biomass production of the root vegetables but not the leafy greens (Fig 4). Among treatments, we found no differences in aboveground biomass of mustard greens (p = 0.09) nor kale (p = 0.83). Beet and turnip plants with buried copper both produced more aboveground biomass than their control plants (both p < 0.01). Compared to exposed copper, buried copper marginally increased beet aboveground biomass (p = 0.054), but did not affect turnip aboveground biomass (p = 0.08). Neither beet nor turnip aboveground biomass differed between exposed copper and control treatments (p = 0.42 and 0.27, respectively). However, control plant biomass was lower than the mean of combined copper treatments (p = 0.01) for both species.

We did not find any effect of copper on belowground biomass production except for turnips with buried copper (Fig 5). Turnip plants with buried copper had greater belowground biomass that the control (p = 0.01) and plants with exposed copper (p = 0.03), and control plants did not differ from exposed copper (0.83) or from the mean of combined copper treatments (0.06). Beet belowground biomass did not differ among treatments (p = 0.44).

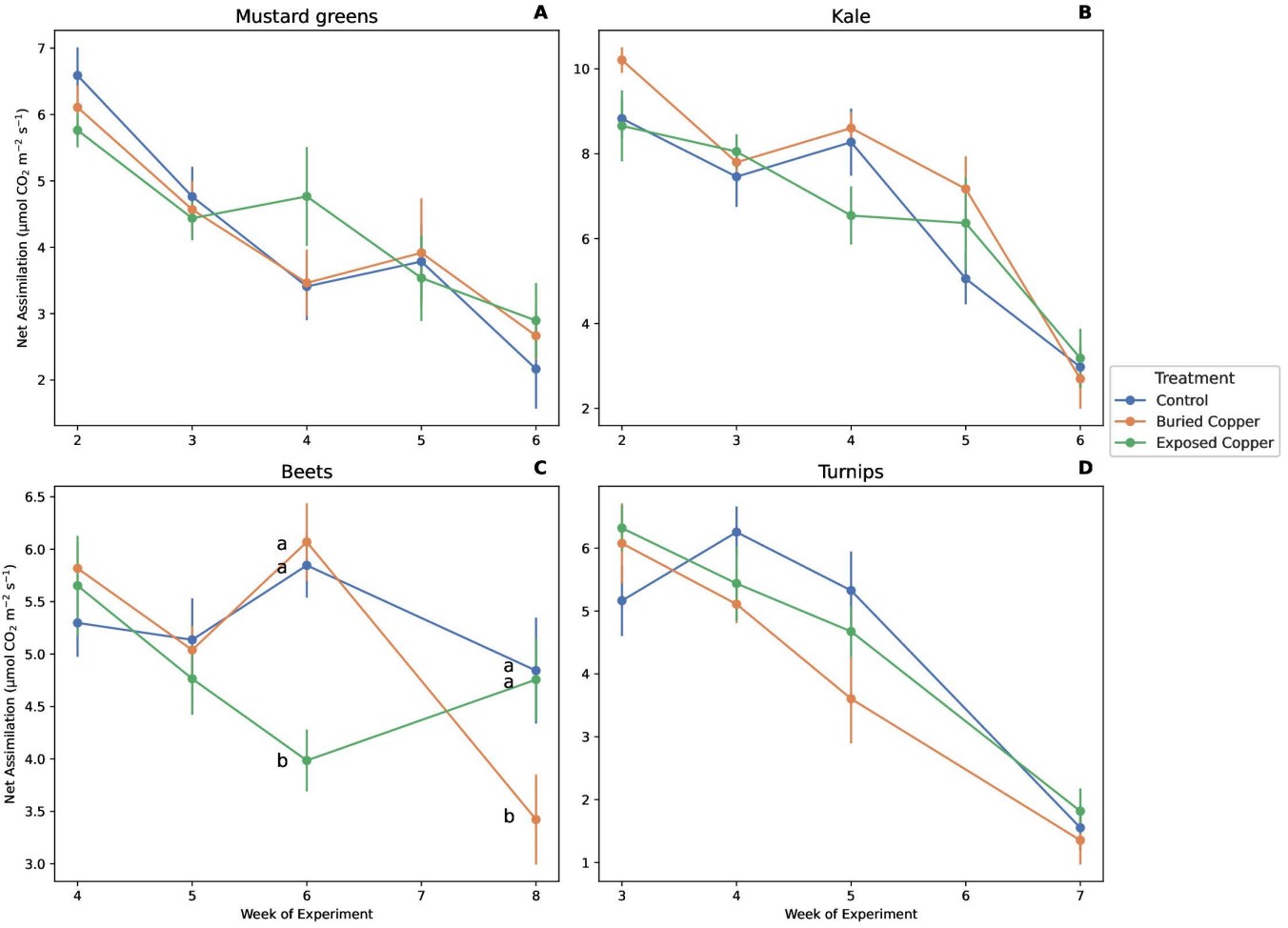

**Fig 1. Effects of week and copper treatment on *Anet* of A) mustard greens, B) kale, C) beets, and D) turnips.** Points show mean ± 1 SE. Letters denote significant differences within weeks when a treatment x interaction was significant.

## Discussion

In this study, we tested the hypotheses that copper-wrapped dowel rods would not improve plant physiology, but if they did then the effect would be attributed to copper fertilization rather than passive electrical stimulation. We found no consistent evidence that passive electroculture is beneficial to crop growth or yield in container gardens. Although we documented statistically significant effects of buried copper on the above- and belowground biomass of turnips, it is unlikely that improved turnip yield was due to copper fertilization because plants grown with exposed copper rods did not show the same effect.

Several centuries of research have demonstrated that crop growth and yield are improved by electrical stimulation [2,5,6]. However, electrical voltages range between 6–1200 V, with most tests using over 200 V to assess the effectiveness of electroculture [14]. In fact, one study found that application of a 40,000 V electrical field improved wheat yields [22]. The lowest tested voltage reported in [14], 6 V, increased both above and belowground biomass of tomato plants [23]. Weaker, pulsed electrical fields can promote the production and retention of antioxidant compounds, such as vitamin C (ascorbic acid) and catalase [24,25], and other secondary metabolites [26]. These results suggest that the application

**Table 1. Model comparison results for repeated measures analysis of height, SPAD, $A_{net}$, and $g_{sw}$ for each of the four species (*: marginally significant, **: statistically significant).**

| | Mustard Greens | | Kale | | | Beets | | | Turnips | |
|---|---|---|---|---|---|---|---|---|---|---|
| **Height** | $\chi^2$ | $p$ | $\chi^2$ | $p$ | | $\chi^2$ | $p$ | | $\chi^2$ | $p$ |
| week*treatment | 17.11 | 0.07 | 9.15 | 0.52 | | 20.75 | 0.05 | * | 6.08 | 0.81 |
| week+treatment | 2.11 | 0.35 | 3.39 | 0.18 | | 0.93 | 0.63 | | 4.37 | 0.11 |
| **SPAD** | | | | | | | | | | |
| week*treatment | 4.04 | 0.95 | 4.04 | 0.95 | | 10.25 | 0.42 | | 5.97 | 0.82 |
| week+treatment | 4.48 | 0.11 | 0.42 | 0.81 | | 2.20 | 0.33 | | 2.54 | 0.28 |
| **$A_{net}$** | | | | | | | | | | |
| week*treatment | 2.36 | 0.97 | 9.96 | 0.27 | | **20.42** | **0.00** | ** | 8.90 | 0.18 |
| week+treatment | 0.46 | 0.79 | 5.02 | 0.08 | | **8.66** | **0.01** | ** | 2.84 | 0.24 |
| **$g_{sw}$** | | | | | | | | | | |
| week*treatment | 0.44 | 1 | 13.97 | 0.27 | | 13.06 | 0.11 | | 11.66 | 0.17 |
| week+treatment | 2.63 | 0.27 | **9.87** | **0.01** | ** | 4.18 | 0.12 | | 3.27 | 0.20 |

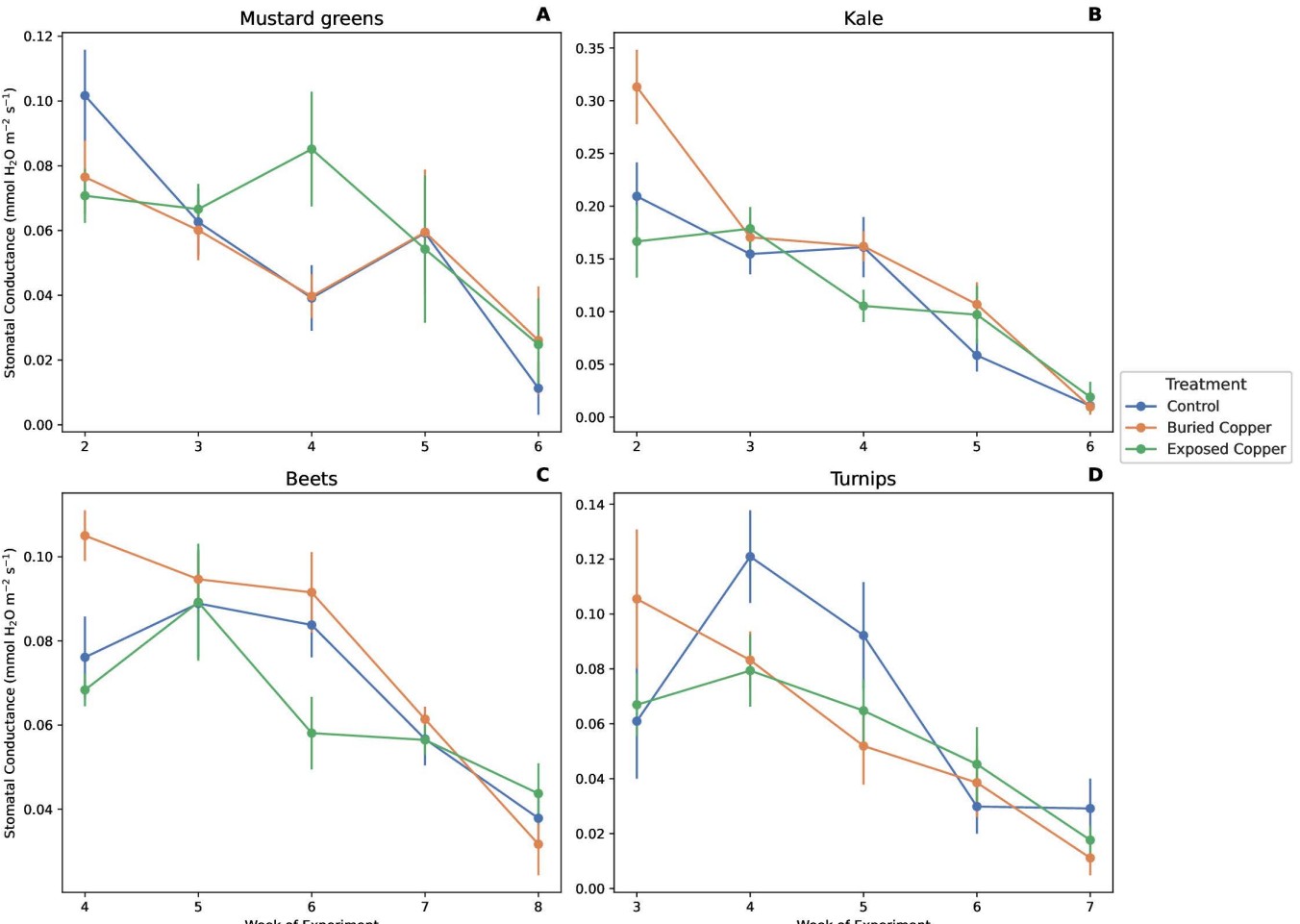

**Fig 2. Effects of week and copper treatment on *gsw* of A) mustard greens, B) kale, C) beets, and D) turnips.** Points show mean ± 1 SE. Letters denote significant differences within weeks when a treatment x interaction was significant.

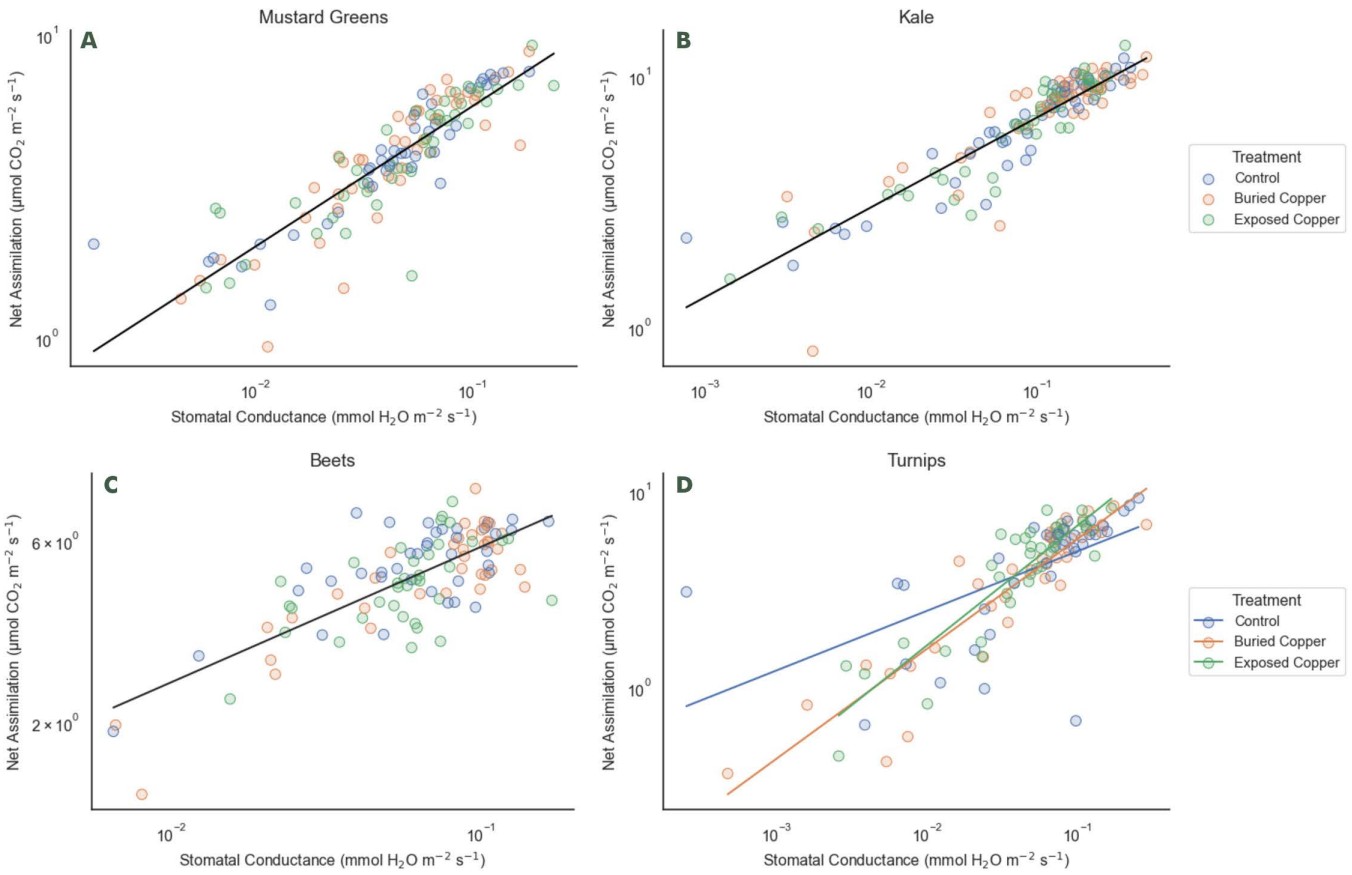

**Fig 3. Linear regressions between *Anet* and *gsw*, representing water user efficiency (WUE), for each of A) mustard greens, B) kale, C) beets, and D) turnips.**

**Table 2. ANCOVA tables for the relationships between $A_{net}$ and $g_{sw}$ for each of mustard greens, kale, beets, and turnips.**

|  | Mustard Greens | | Kale | |
|---|---|---|---|---|
|  | Fvalue | Pvalue | Fvalue | Pvalue |
| log(gsw) | 431.81 | <0.001 | 610.68 | <0.001 |
| treatment | 0.224 | 0.799 | 0.065 | 0.937 |
| log(gsw):treatment | 1.774 | 0.174 | 0.556 | 0.575 |
|  | Beets | | Turnips | |
|  | Fvalue | Pvalue | Fvalue | Pvalue |
| log(gsw) | 143.19 | <0.001 | 294.57 | <0.001 |
| treatment | 2.294 | 0.106 | 1.303 | 0.276 |
| log(gsw):treatment | 2.188 | 0.117 | 0.893 | 0.413 |

of even weak electrical fields can potentially improve plant yield, stress tolerance, or herbivore resistance. However, it is unlikely that passive electrical transport of copper rods provides enough voltage affect plants. Electrical transmittance to the soil requires a voltage differential between the soil and the atmosphere. The commonly accepted differential is 100 V per meter, such that the voltage differential between the soil and the top of a 40 cm copper rod is only 4 V. We measured

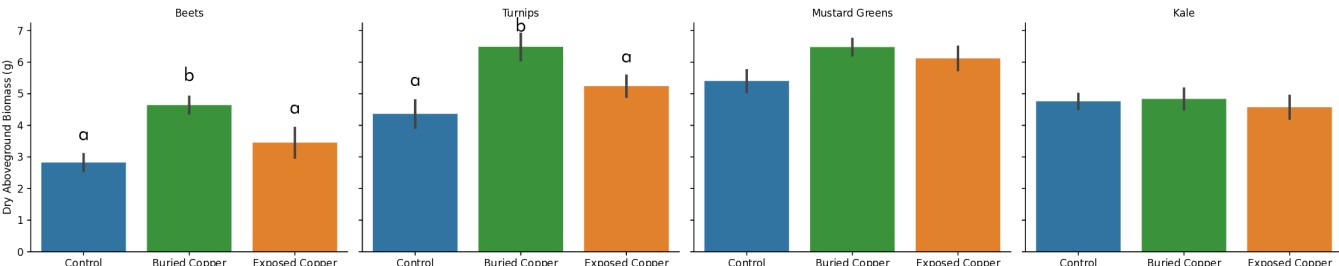

**Fig 4. Bar charts of aboveground biomass production for beets, turnips, mustard greens, and kale.** Bars show mean +/- 1 SEM. Letters denote statistically different groups.

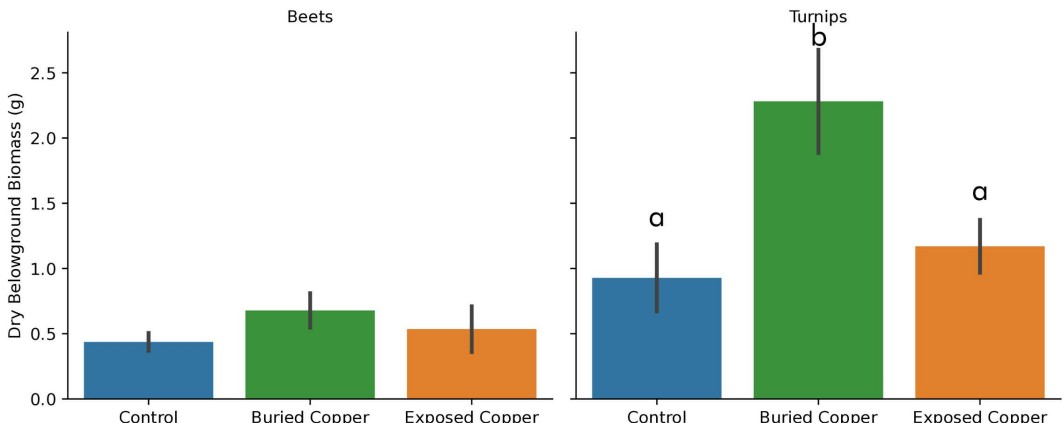

**Fig 5. Bar charts of belowground biomass production for beets and turnips.** Bars show mean +/- 1 SEM. Letters denote statistically different groups.

electrical transmission directly using an oscilloscope and found that copper rods placed randomly around the room transmit only 2 millivolts into the soil. When placed within 4 cm of a fluorescent light, transmittance increased to a maximum of 20 millivolts. Thus, copper rods do transmit electricity into the soil but at voltages that are unlikely to have any impact on plant physiology or performance.

Net $CO_2$ assimilation of beets did seem to vary by copper treatment, with exposed copper having significantly lower $A$ in week 6 and buried copper having significantly reduced $A$ during week 8. Given the inconsistent results, these patterns are not due to copper treatment *per se.* Instead, both low measurements are likely experimental artifacts. We rotated plants around our growth table on a weekly basis to minimize the potential confounding effects of light or other small variations in the room. Both abnormally low measurements (week 6 exposed copper, week 8 buried copper) occurred in Block C. It is most likely that this block received lower light levels than other blocks at the time of measurement and therefore affected $A$, rather than the copper treatment. Furthermore, soil moisture in the buried copper treatment for beets was ~15% compared to 17.5% in the other two treatments, further evidence that experimental differences account for the observed patterns rather than any treatment effect.

We also hypothesized that the insertion of copper rods into the soil might provide fertilization benefit. At low concentrations, copper fertilization can improve crop yield in copper-limited soils [18–20]. However, many potting soils for home container use contain copper sulfate as an anti-bacterial and anti-fungal agent, likely providing plants with adequate copper. Naturally, soil copper concentrations in both the US and Europe [27,28] are sufficient to alleviate copper limitation, with

potential exceptions in the southeast US, western Michigan, and eastern Europe and Scandanavia. Indeed, we did find that buried copper rods increased aboveground biomass of both root vegetables, beets and turnips, by ~2.5 g (Fig 4), and also increased belowground turnip yield by 1 g (Fig 5). Yet we cannot conclude that these effects are due to copper fertilization because the exposed copper rods, with a similar amount of copper buried in the soil, did not increase either the above or belowground biomass of root vegetables (Figs 4,5). Soil moisture measures also do not explain this difference; there was no systematic difference in soil moisture among treatments for either beets ($p = 0.219$) or turnips ($p = 0.175$). Thus, as neither copper treatment nor soil moisture can explain why beets and turnips with buried copper rods performed best, and it is therefore likely that this pattern is simply a result of small sample sizes.

It is possible that copper fertilization can improve crop yields in mineral-deficient soils. Spraying 5 kg ha$^{-1}$ of copper improved soybean yields on Indian Mollisol [29], enhanced gas exchange of coconut seedlings in a greenhouse experiment [30], and even the addition of 0.25 kg ha$^{-1}$ improved wheat yield in the copper-deficient soils of Poland [31]. However, it is unlikely that the addition of copper in the form of a solid metal rod will improve growth, even in copper limited soils. Copper likely does not leach from the metal rod fast enough to infiltrate soils and plant roots. Even if leaching occurs, copper is more downwardly mobile than laterally mobile in soils and is likely to wash downward rather than spread laterally through the soils towards the plant [32–34]. Most studies of copper fertilization use a liquid form of copper (*i.e.,* copper sulfate) sprayed directly onto foliar surfaces [29–31], which ensures uptake by plant leaves and roots.

In summary, we have conducted a controlled experiment on multiple crops, including leafy greens and root vegetables, and found no evidence that passive electroculture can improve plant growth, photosynthesis, or yield. While crop production could potentially be enhanced by the application of electrical fields, future work could examine exciting possibilities of using small solar panels to apply a constant, minimal current to soils or even directly to plant surfaces in order to enhance yield in urban settings. Future work should also examine the voltage thresholds required for improving crop yield. However, the economic feasibility of current applications restricts these studies to urban container gardens, but could still provide a boost in food security in urban settings. Unfortunately, the voltages required exceed what is produced by copper-wrapped wooden dowels. We therefore suggest that both the production and purchase of such products would waste both financial and natural resources.

## Acknowledgments

We would like to thank Tom Dunk for his electrical expertise.

## Author contributions

**Conceptualization:** Mya Chier, Nathan Lemoine.

**Data curation:** Michelle L Budny, Nathan Lemoine.

**Formal analysis:** Mya Chier, Michelle L Budny, Nathan Lemoine.

**Funding acquisition:** Nathan Lemoine.

**Investigation:** Mya Chier, Michelle L Budny.

**Methodology:** Mya Chier.

**Project administration:** Nathan Lemoine.

**Supervision:** Nathan Lemoine.

**Visualization:** Mya Chier, Michelle L Budny, Nathan Lemoine.

**Writing – original draft:** Mya Chier, Aidan Oakey, Michelle L Budny, Nathan Lemoine.

**Writing – review & editing:** Aidan Oakey, Michelle L Budny, Nathan Lemoine.

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
