## [Decision Letter · Decision Letter 0]

29 May 2025

PONE-D-25-18809Passive electroculture using copper rods does not improve vegetable yield.PLOS ONE

Dear Dr. Lemoine,

Thank you for submitting your manuscript to PLOS ONE. After careful consideration, we feel that it has merit but does not fully meet PLOS ONE’s publication criteria as it currently stands. Therefore, we invite you to submit a revised version of the manuscript that addresses the points raised during the review process.

We look forward to receiving your revised manuscript.

Kind regards,

Debasis Mitra

Academic Editor

PLOS ONE

Journal Requirements:

 [NSF DEB 1941390].

Additional Editor Comments (if provided):

Reviewers' comments:

Reviewer's Responses to Questions

**Comments to the Author**

1. Is the manuscript technically sound, and do the data support the conclusions?

Reviewer #1: Partly

Reviewer #2: Yes

2. Has the statistical analysis been performed appropriately and rigorously? 

Reviewer #1: Yes

Reviewer #2: Yes

3. Have the authors made all data underlying the findings in their manuscript fully available?

Reviewer #1: Yes

Reviewer #2: Yes

4. Is the manuscript presented in an intelligible fashion and written in standard English?

Reviewer #1: Yes

Reviewer #2: Yes

5. Review Comments to the Author

Reviewer #1: This manuscript evaluates the effects of passive electroculture using copper-wrapped dowels on plant growth and yield. While the study is timely and provides useful data to assess popular gardening claims, several key issues need to be addressed, particularly regarding terminology, interpretation, and discussion of electrical mechanisms. Major revision is needed before it can be considered for publication. Detailed comments are provided below.

1. The title "Passive electroculture using copper rods does not improve vegetable yield" is too definitive given the limited scope and specific conditions of the study.

2. While the study addresses a timely and socially relevant topic, its contribution to advancing fundamental scientific understanding is limited. The work is well suited to debunk pseudoscience but lacks sufficient mechanistic insight to support publication as a scientific article without further strengthening.

3. The manuscript frequently refers to the intervention as “electroculture,” yet no actual electric current or active field was applied. This is conceptually misleading. Suggest to add “passive” before electroculture.

4. The interpretation of differential effects between buried and exposed copper rods is not well supported. Since both treatments include similar amounts of copper in the soil, and the exposure difference is minimal in terms of electrical conduction, the observed differences in plant response remain unclear. Additional discussion and, if possible, data (e.g., copper ion concentrations or soil redox status) would improve clarity.

5. Despite the manuscript’s framing around “electroculture,” the discussion of electrical mechanisms is limited to only a few sentences and includes no in-depth review of relevant literature on voltage thresholds or field strength in passive systems. Given the central theme of electroculture, the authors should substantially expand this section by including additional studies and theoretical explanations about how passive systems could (or could not) influence plant physiology.

Reviewer #2: The work mentioned in the manuscript is scientifically executed and the manuscript is written in proper English. The recommendations to be added: 1. If the study is done in copper deficient soil, will the same result be reciprocated? Please explain this with proper scientific proof or citations. 2. The future scope is not totally clear for future researchers. Please chalk out the future outlook in proper manner.

6. PLOS authors have the option to publish the peer review history of their article (what does this mean? ). If published, this will include your full peer review and any attached files.

**Do you want your identity to be public for this peer review?** For information about this choice, including consent withdrawal, please see our Privacy Policy .

Reviewer #1: **Yes: ** Mairui Zhang

Reviewer #2: **Yes: ** Biswajit Pramanik

---

## [Author Response · Author response to Decision Letter 1]

23 Jun 2025

Reviewer #1

This manuscript evaluates the effects of passive electroculture using copper-wrapped dowels on plant growth and yield. While the study is timely and provides useful data to assess popular gardening claims, several key issues need to be addressed, particularly regarding terminology, interpretation, and discussion of electrical mechanisms. Major revision is needed before it can be considered for publication. Detailed comments are provided below.

1. The title "Passive electroculture using copper rods does not improve vegetable yield" is too definitive given the limited scope and specific conditions of the study.

We thank the reviewer for this suggestion, because the phrase ‘vegetable yield’ does imply that our results could apply to larger scale agricultural systems or to outdoor home gardens. We have therefore changed our title to “Passive electroculture using copper rods does not improve vegetable yield in home container gardens” to further limit the scope to home (i.e. non-commercial) container (i.e. non-outdoor) vegetable gardening, as this is where most electroculture social media is concentrated. We have left the word ‘vegetable’ because we were broad and deliberate in our selection of two leafy green and two root vegetable species, such that our results are generally applicable to multiple vegetable types.

2. While the study addresses a timely and socially relevant topic, its contribution to advancing fundamental scientific understanding is limited. The work is well suited to debunk pseudoscience but lacks sufficient mechanistic insight to support publication as a scientific article without further strengthening.

The reviewer is correct in that our study does not advance fundamental scientific understanding, because the entire purpose of this experiment and publication is to debunk pseudoscience. Such publications are necessary despite their lack of advancement of fundamental knowledge because pseudoscience is currently all-too-prominent, leading to wasteful spending and, in some cases, endangerment of human health. Rigorous studies that debunk specious claims are therefore worthwhile in and of themselves. The reviewer is also welcome to suggest that this manuscript lacks mechanistic insight and requires further strengthening, but without further explanation as to what particularly could be strengthened, we are at a loss as to how to address such a vague and general comment. Moreover, the mission statement of PloS One is “We evaluate research on the basis of scientific validity, strong methodology, and high ethical standards—not perceived significance.” That is, unless there are fundamental flaws in either the methodology or ethics (which the reviewer does not report), the judgment of whether this manuscript advances fundamental scientific understanding (i.e. our study’s perceived significance) is not a valid critique for the mission of PloS One.

3. The manuscript frequently refers to the intervention as “electroculture,” yet no actual electric current or active field was applied. This is conceptually misleading. Suggest to add “passive” before electroculture.

We apologize for this oversight. Our original experimental design did have an active electrical current treatment, but our electrical probes corroded during the experiment and we were unable to include our results pertaining to the active electroculture. We have followed the reviewer’s advice and ensured that we refer to electroculture as ‘passive’, where appropriate, throughout the manuscript. We found two instances that required clarification on Line 102 and Line 183. Fortunately, most of our references to electroculture in the hypotheses, results, and discussion either already specified our focus on passive electroculture or referred to ‘copper treatments’ that did not mention electroculture. We hope our corrections now sufficiently ensures that our treatment is passive.

4. The interpretation of differential effects between buried and exposed copper rods is not well supported. Since both treatments include similar amounts of copper in the soil, and the exposure difference is minimal in terms of electrical conduction, the observed differences in plant response remain unclear. Additional discussion and, if possible, data (e.g., copper ion concentrations or soil redox status) would improve clarity.

Thank you for pointing this out. We do not have copper ion concentrations nor soil redox status. However, we have been able to determine what we think are at least likely contributing factors to the sporadic effects of copper:

“Net CO2 assimilation of beets did seem to vary by copper treatment, with exposed copper having significantly lower A in week 6 and buried copper having significantly reduced A during week 8. Given the inconsistent results, these patterns are not due to copper treatment per se. Instead, both low measurements are likely experimental artifacts. We rotated plants around our growth table on a weekly basis to minimize the potential confounding effects of light or other small variations in the room. Both abnormally low measurements (week 6 exposed copper, week 8 buried copper) occurred in Block C. It is most likely that this block received lower light levels than other blocks at the time of measurement and therefore affected A, rather than the copper treatment. Furthermore, soil moisture in the buried copper treatment for beets was ~15% compared to 17.5% in the other two treatments, further evidence that experimental differences account for the observed patterns rather than any treatment effect.” (lines 224 - 234)

and

“Soil moisture measures also do not explain this difference; there was no systematic difference in soil moisture among treatments for either beets (p = 0.219) or turnips (p = 0.175). Thus, as neither copper treatment nor soil moisture can explain why beets and turnips with buried copper rods performed best, and it is therefore likely that this pattern is simply a result of small sample sizes.” (lines 248 - 252)

5. Despite the manuscript’s framing around “electroculture,” the discussion of electrical mechanisms is limited to only a few sentences and includes no in-depth review of relevant literature on voltage thresholds or field strength in passive systems. Given the central theme of electroculture, the authors should substantially expand this section by including additional studies and theoretical explanations about how passive systems could (or could not) influence plant physiology.

The reviewer is correct in that we could have expanded our discussion of electrical mechanisms. Unfortunately, our study is focused on the passive electroculture provided by copper rods, and a Web of Science search of ‘passive electroculture’ returned no results. Thus, we must restrict our disucssion to the relatively limited amount of data on weak pulsed electrical fields. While our Introduction has already discussed the idea that electrical signaling is an important trigger of many plant pathways that could be manipulated by electrical application, we have also added in new lines to the Discussion:

“Weaker, pulsed electrical fields can promote the production and retention of antioxidant compounds, such as vitamin C (ascorbic acid) and catalase (Radhakrishnan and Kumari, Silva-Fortuny et al.), and other secondary metabolites (Ye et al. 2004). These results suggest that the application of even weak electrical fields can potentially improve plant yield, stress tolerance, or herbivore resistance.” (lines 193-196)

Unfortunately, there is simply not much more information to add about weak electrical fields in general, and none on passive systems like copper rods.

Reviewer #2

The work mentioned in the manuscript is scientifically executed and the manuscript is written in proper English. The recommendations to be added:

1. If the study is done in copper deficient soil, will the same result be reciprocated? Please explain this with proper scientific proof or citations.

This is a good suggestion. We have added in a discussion:

“It is possible that copper fertilization can improve crop yields in mineral-deficient soils. Spraying 5 kg ha-1 of copper improved soybean yields on Indian Mollisol [26], enhanced gas exchange of coconut seedlings in a greenhouse experiment [27], and even the addition of 0.25 kg ha-1 improved wheat yield in the copper-deficient soils of Poland [28]. However, it is unlikely that the addition of copper in the form of a solid metal rod will improve growth, even in copper limit soils. Copper likely does not leach from the metal rod fast enough to infiltrate soils and plant roots. Even if leaching occurs, copper is more downwardly mobile than laterally mobile in soils and is likely to wash downward rather than spread laterally through the soils towards the plant [29 – 31]. Most studies of copper fertilization use a liquid form of copper (i.e. copper sulfate) sprayed directly onto foliar surfaces [26 – 28], which ensures uptake by plant leaves and roots.” (lines 235 - 244)

2. The future scope is not totally clear for future researchers. Please chalk out the future outlook in proper manner.

We have added in a more direct statement of future work:

“While crop production could potentially be enhanced by the application of electrical fields, future work could examine exciting possibilities of using small solar panels to apply a constant, minimal current to soils or even directly to plant surfaces in order to enhance yield in urban settings. Future work should also examine the voltage thresholds required for improving crop yield. However, the economic feasibility of current applications restricts these studies to urban container gardens, but could still provide a boost in food security in urban settings.” (lines 248 - 254).

---

## [Decision Letter · Decision Letter 1]

21 Jul 2025

Passive Electroculture Using Copper Rods Does Not Improve Yield in Home Container Vegetable Gardening

PONE-D-25-18809R1

Dear Dr. Lemoine,

We’re pleased to inform you that your manuscript has been judged scientifically suitable for publication and will be formally accepted for publication once it meets all outstanding technical requirements.

Kind regards,

Debasis Mitra

Academic Editor

PLOS ONE

Additional Editor Comments (optional):

Reviewers' comments:

Reviewer #1: (No Response)

Reviewer #2: Mostly each comments have been properly addressed by the authors and thus, I recommend to accept this manuscript to be published under the esteemed journal.

---

## [Editor Report · Acceptance letter]

PONE-D-25-18809R1

PLOS ONE

Dear Dr. Lemoine,

I'm pleased to inform you that your manuscript has been deemed suitable for publication in PLOS ONE. Congratulations! Your manuscript is now being handed over to our production team.

Kind regards,

on behalf of

Dr. Debasis Mitra

Academic Editor

PLOS ONE